# Evaluating socio-economic and conservation impacts of management: A case study of time-area closures on Georges Bank

**David M. Keith** [1,2]*, **Jessica A. Sameoto**[1], **Freya M. Keyser**[1], **Christine A. Ward-Paige**[3]

**1** Population Ecology Division, Fisheries and Oceans Canada, Bedford Institute of Oceanography, Dartmouth, Nova Scotia, Canada, **2** Department of Biology, Dalhousie University, Halifax, Nova Scotia, Canada, **3** eOceans, Dartmouth, Nova Scotia, Canada

* david.keith@dfo-mpo.gc.ca

**Data Availability Statement:** The majority of the data used in this manuscript are subject to legal restrictions. The Canadian Privacy Act restricts our ability to share the original commercial fishery data

## Abstract

Globally, economies and marine ecosystems are increasingly dependent on sustainable fisheries management (SFM) to balance social, economic, and conservation needs. The overarching objectives of SFM are to maximize both conservation and socio-economic benefits, while minimizing short-term socio-economic costs. A number of tools have been developed to achieve SFM objectives, ranging from fishery specific to ecosystem-based strategies. Closures are a common SFM tool used to balance the trade-off between socio-economic and conservation considerations; they vary in scope from small-scale temporary closures to large-scale permanent networks. Unfortunately, closures are frequently implemented without a plan for monitoring or assessing whether SFM objectives are met. In situations in which a monitoring plan is not in place we propose that commonly available fishery data can often be used to evaluate whether management tools are effective in meeting SFM objectives. Here, we present a case study of closures on Georges Bank that shows how fishery data can be analyzed to perform such an assessment. Since 2006, on the Canadian side of Georges Bank, seasonal scallop fishery closures have been implemented with the aim of reducing by-catch of Atlantic cod (*Gadus morhua*) and yellowtail flounder (*Pleuronectes ferruginea*) during spawning. In lieu of data from a dedicated monitoring program, we analyzed data from Vessel Monitoring Systems (VMS), fishery logbooks, and a scallop survey to assess the impact of these closures on the scallop fishery, and use observer data (i.e. by-catch) to assess the effectiveness of these closures in meeting their conservation objective. While compliance for these time-area closures was high, the closures did not significantly displace fishing activity and overall there was limited evidence of an impact on the scallop fishery. Further, the discard rates for both cod and yellowtail were above average when their respective closures were active. These results suggest that improvements to the closures design and/or other measures may be required to achieve the desired SFM objectives.

used in this analysis. This act sets out rules for how institutions of the Government of Canada must deal with personal information of individuals. These data are considered private third party information and are subject to privacy restrictions; this includes logbook data, VMS data, and observer data. Commercial data are house and archived by Fisheries and Oceans Canada (DFO) and any request for these commercial data would be through the DFO Maritimes Region Commercial Data Division and would be subject to review by the Access to Information and Privacy Secretariat within DFO. In some cases it is possible that data could be released if these data could be rendered anonymous through aggregation. The scallop biomass information is collected through an industry funded survey through a legal agreement with DFO; these data are stored and maintained by DFO and access may be requested through DFO. Links to DFO data contacts are in attached cover letter and below. Fishery Data: (email) Email: XMARComData@dfo-mpo.gc.ca Scallop survey biomass (email): DFO.MAR-PED-Data-Request-Demande-de-donnes-DEP-MAR.MPO@dfo-mpo.gc.ca.

**Funding:** The author(s) received no specific funding for this work.

# Introduction

While humans have relied on fisheries for centuries, population increases and technological advances have resulted in increased pressures on both fisheries and marine ecosystems [1]. Sustainability strategies have been developed with the aim of balancing trade-offs between these growing socio-economic pressures and ecosystem conservation [2, 3]. In the marine realm, sustainability strategies primarily focus on fisheries, vulnerable species and ecosystems, and the protection of marine habitats (e.g. Aichi Biodiversity targets 6, 10, and 11). Typically, sustainable fisheries management (SFM) strategies seek to optimize long-term socio-economic benefits (e.g. economic yield or employment) in part by ensuring that populations are maintained above biological reference levels [4].

A suite of SFM tools have been developed in an effort to manage fisheries and marine ecosystems. These tools include fishery specific controls that limit the number of vessels, days at sea, by-catch, and the total allowable catch, as well as tools that focus more directly on the ecosystem [5, 6]. Ecosystem based SFM tools often involve limiting some forms of human use to protect vulnerable species, unique habitats, or ecosystems. Closures, which are a commonly employed management tool, are used to limit access to an area and can range in scope from small temporary area closures directed at limiting one specific activity to networks of large permanent marine reserves that prohibit a wide range of activities [7–10].

The scope and type of SFM tool implemented is often influenced by both socio-economic and conservation objectives [11–13]. Closure tools include; a) Marine Protected Areas (MPAs), which are permanent closures and can exclude a range of activities, including fishing that may obstruct the MPA objectives [7], b) protection of sensitive benthic species and habitats by the exclusion of certain fishing activities [14], c) time-area closures, which temporarily restrict fishing activity to protect a species during a vulnerable life-history stage [15, 16], and d) dynamic closures, which can vary spatio-temporally at differing scales using near real-time data [17–19]. The overarching objective of any of these closure tools is to maximize the conservation benefit while minimizing the socio-economic impact [2]. MPAs (a) are rarely implemented without extensive consultation with stakeholders and socio-economic considerations are explicitly accounted for in their development [2, 11]. The more targeted closure tools (b-d) are typically implemented in cases where broad fisheries restrictions are deemed unacceptable for socio-economic reasons and specific, selective management action(s) is preferred to achieve conservation objectives.

The conservation and socio-economic impacts of closures have been debated for decades [20–24] and a major challenge for many closures is to quantify these impacts [25, 26]. This is especially problematic if a proper monitoring strategy is not implemented [27]. The closure design process should identify a) the SFM objectives of the closure, b) the indicators required to quantify if these objectives are being achieved, and c) a monitoring strategy to measure and monitor the chosen indicators [28, 29]. Without a monitoring plan, it is difficult to evaluate whether the closure is effective and achieving the SFM objectives [10, 28]. Further, if not designed carefully, the closure may be ineffective or may have net-negative impacts from a socio-economic and/or conservation perspective [10, 18, 30].

These concerns are amplified for closures in which only a portion of an area is closed for part of the year. A *time-area closure* must be designed at the appropriate spatial and temporal scale to increase the probability of meeting both socio-economic and conservation objectives [31–33]. Time-area closures can result in the displacement of fishing effort to lower quality habitat outside the closure area. In these cases, if the spatial domain of the closure does not encompass all of the components of the stock targeted for protection, the exploitation experienced by the stock that the closure was designed to protect may increase [34–36] and neither

the socio-economic nor conservation objectives may be achieved. Similarly, a closure with a duration that is shorter than the time-frame of the behaviour targeted for protection (e.g. spawning) can also lead to changes in fishery behaviour, which results in elevated exploitation after the closure reopens [37] and increased fishing effort during non-optimal periods. If the implementation of the closure also does not account for potential inter-annual variability (e.g. oceanographic conditions) or long-term shifts in life histories (e.g. spawning timing changes due to ocean warming), the closure may not actually protect the stock during the life history stage as intended. Furthermore, the size of the closed area must be large enough to encompass the component of the stock identified for protection throughout the duration of the closure [34, 35, 37–39] and this requires a detailed knowledge of the biology of the species and an understanding of the current status of the stock [40]. In situations where a directed monitoring strategy for a time-area closure has not been established, it is difficult to determine the appropriate scale (in both space and time) of closure that would meet the SFM objectives [37, 41]. If the scale is too large, the closure could have detrimental socio-economic impacts, while if the scale is too small, the conservation objectives would not be achieved.

In cases where there is no directed monitoring strategy there may still be a wealth of information collected on a regular basis which can be used to help monitor a closure. This information can include logbook records of fishery location, landings and effort, Vessel Monitoring System (VMS) or Automatic Identification System (AIS) data, observer records, fishery independent data (e.g. species-specific and ecosystem surveys), traditional and local ecological knowledge, and data from citizen science initiatives. These data streams can be used to look for evidence that a closure has impacted the fishery and to determine if the interactions between the fishery and the species of interest have changed [37]. A lack of a measurable impact on the fishery would suggest that the closure has not had a significant socio-economic impact. Conversely, if the interactions between the fishery and the species of interest are not impacted by the closure this would suggest that the conservation objectives of this strategy were not achieved. A successful SFM strategy must have an acceptable socio-economic impact and a significant conservation impact [2].

Several formerly commercially valuable groundfish species on Georges Bank are currently well below mean historical population levels [42, 43]. Two of these stocks, Atlantic cod (*Gadus morhua*) and yellowtail flounder (*Pleuronectes ferruginea*), have supported relatively large fisheries in the past. On the Canadian portion of Georges Bank there currently is no directed fishery for Atlantic cod or yellowtail flounder, however, there are annual by-catch reserves allocated to the Canadian groundfish fishery and the Canadian offshore scallop fishery (hereafter referred to as the scallop fishery). In addition, Fisheries and Oceans Canada (DFO) has implemented time-area closures on Georges Bank; the conservation objectives of these closures are to protect aggregations of Atlantic cod (cod closure) and yellowtail flounder (yellowtail closure) during peak spawning periods from the mobile bottom gear deployed by the scallop fishery.

Here we demonstrate an approach that uses readily available fishery data to assess the socio-economic (fishery) impact of the closures and to evaluate the effectiveness of the closures in achieving their conservation objectives. To evaluate the socio-economic impact of the closure on the scallop fishery three questions were addressed; 1) how do patterns of scallop productivity influence the scallop fishery and did the implementation of the closures influence this relationship?, 2) do the closures displace the scallop fishery from areas that would be fished?, and 3) have the closures impacted the ability of the fishery to reach their total allowable catch (TAC) of scallop? To evaluate whether the conservation objectives of the closures are being achieved, the seasonal patterns of discards from the fishery during the closure era (2006-2018) were analyzed. Unfortunately, there were insufficient observer data from the pre-closure

era (2000-2005) to perform a direct comparison of discards between the pre-closure and clo-sure eras. We discuss the implications of these results for this case study and for SFM more broadly.

## Methods

### Study area

Georges Bank, located in the Northwest Atlantic straddling the United States (U.S.)-Canada maritime border, is a 3-150 m deep plateau that covers approximately 40,000 $km^2$, is character-ized by high primary productivity, and historically high fish abundance [44]. It is an eroding bank with no sediment recharge, and covered with coarse gravel and sand that provides important habitat for many species [45]. Since 1984, Georges Bank has been divided between the U.S. and Canada and, while some collaborative fishery management exists, the U.S. and Canadian portions are largely managed separately (Fig 1).

### The closures

The scallop fishery time-area closures are designed to protect cod and yellowtail at peak spawning times on Georges Bank [46, 47]. The timing of the closures differ by species and are selected based on the respective species' life histories [48]. The location of the closures can vary from year to year, and the final location for a given year is determined by an analysis of fishery and/or survey data [46, 47] followed by consultation with the scallop fishery.

  To determine potential closure areas, Georges Bank is subdivided into a predetermined grid of candidate cells of $\approx 43 \ km^2$ and this grid is used for both closures. For the cod closures, DFO research vessel groundfish survey data is used and a 10-year moving window of cod abundance is calculated to identify cells expected to have the highest cod abundance. Areas classified as 'high abundance' are those with an average of $> 3.5$ standardized individuals. The top cells are then combined with the previous years' first quarter scallop fishery catch, to describe overlap between the two features [47]. The yellowtail closures use groundfish (otter trawl) fishery discard rates and scallop catch in the second quarter of the previous year to iden-tify candidate cells to close and this process was conducted from 2007 to 2014; however, the location of the yellowtail closures has remained static since 2014 [46].

  DFO Resource Management uses the most recent science advice to determine the cells that should be temporarily closed each year. Both closures have consisted of a series of 3 to 7 cells in each year (Table 1). Closed cells have comprised a total closed area of 128 to 300 $km^2$ per year, or 6280 $km^2$ across all years. Atlantic cod closures ranged from 37 to 58 days in February to the end of March, and yellowtail flounder closures have always been for the duration of June.

### The fishery

The scallop fishery currently fishes up to seven banks in Canadian waters (Fig 1). These banks occur within scallop fishing areas that have a discrete total allowable catch (TAC) that is set by DFO. The fishery predominately lands only the scallop adductor muscle (the 'meat') and it is the total weight of the meat landed that contributes to the TAC. The scallop fleet deploys ben-thic dredges to fish year round from January to December using two types of vessels that vary in catch rates and processing methods. Atlantic sea scallop (*Placopecten magellanicus*) on Georges Bank were previously fished to low levels in the 1980s but the fishery is now consid-ered a sustainable and economically valuable fishery [49]. The primary scallop fishing area on Georges Bank covers approximately 4250 $km^2$ (Fig 1). The Canadian portion of Georges Bank

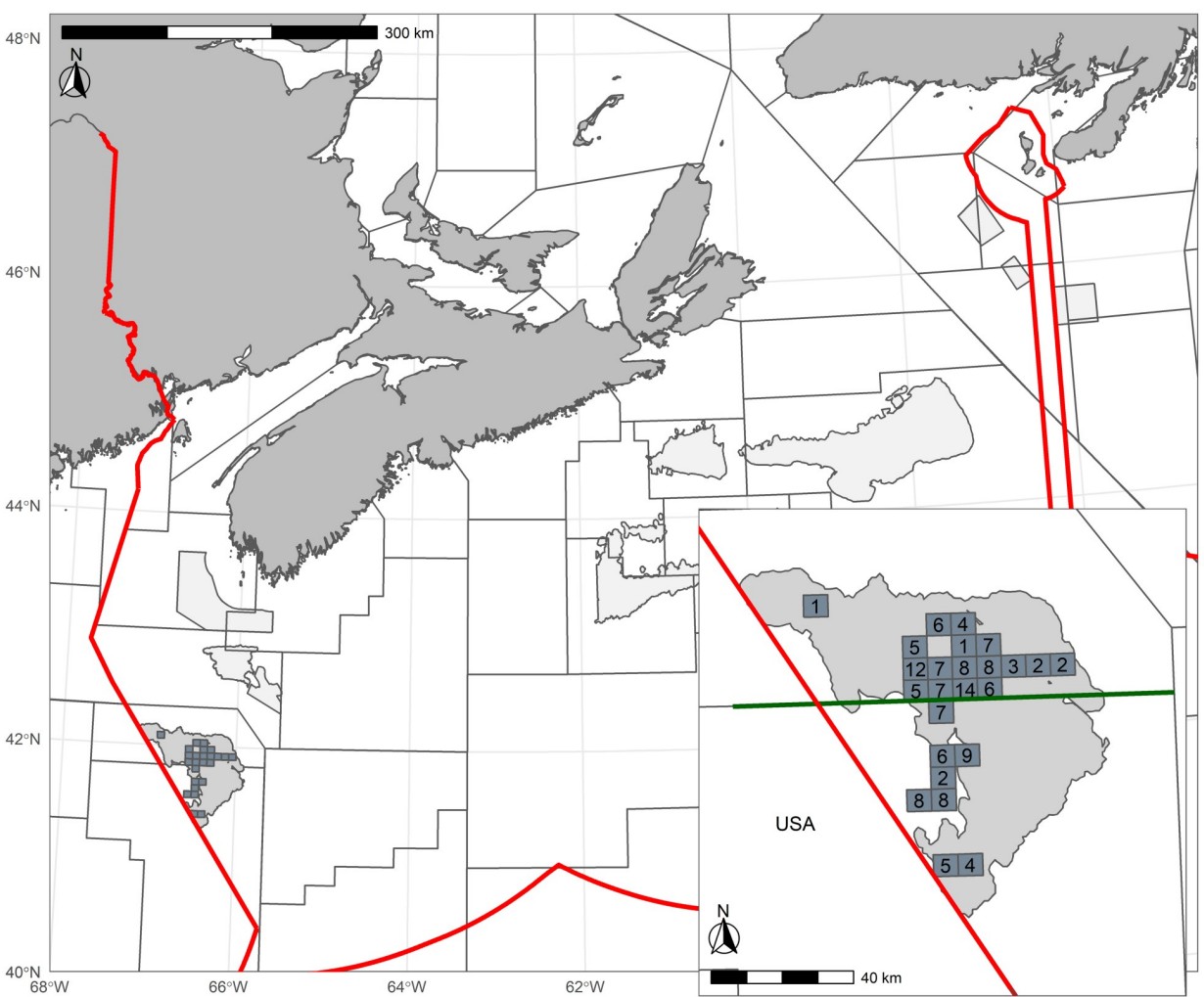

**Fig 1. Map of the study area.** The light grey polygons indicate the primary scallop banks which are regularly fished by the offshore scallop fishery. In the inset the grey polygon is the main study area on Georges Bank and the darker grey polygons indicate a cell that has been closed at least once since 2006. The green line represents the north-south division of the fishing area on Georges Bank. The numbers inside the cells indicate the total number of times a cell has been closed. While relatively infrequent, a given cell can be closed twice in a single year, once during the cod closure and once during the yellowtail closure. Closures for cod occurred in February-March, 2006-2018; yellowtail flounder closures occurred in June, 2007-2018. Red lines indicate the Canadian exclusive economic zone boundary.

contains two NAFO sub-regions (5ZEm and 5ZEj). These sub-regions happen to delineate a northern region of higher scallop productivity from a less productive southern region (Fig 1 inset).

Logbooks are required by DFO and the records from the scallop fishery from 2000 until 2018 were used in this study; these data include, the date, location, effort, and catch. The records were daily until 2008; beginning in 2009 the logbooks have been completed every 6 hours. The catch data is entered in terms of the estimated meat weight caught for the time period covered by each logbook record. Upon completion of a trip the landings are inspected and weighed by dockside monitors. Since logbook records contain estimates of catch, the catch estimate for each logbook entry is prorated as a function of the proportion of estimated catch to total trip landings; the prorated estimated catch for each logbook record was subsequently used.

**Table 1. Summary of the past closures.** Area is in $km^2$, perimeter is $km$. The aggregation index is the ratio of the perimeter to area, lower values indicate increased aggregation. Each cell has an area of $\approx 43 km^2$.

| Closure | Year | Days | Area | Perimeter | Aggregation index |
|---|---|---|---|---|---|
| Cod | 2006 | 46 | 213 [$km^2$] | 89 [km] | 0.42 [1/km] |
| | 2007 | 37 | 256 [$km^2$] | 120 [km] | 0.45 [1/km] |
| | 2008 | 51 | 299 [$km^2$] | 120 [km] | 0.39 [1/km] |
| | 2009 | 51 | 299 [$km^2$] | 120 [km] | 0.39 [1/km] |
| | 2010 | 51 | 299 [$km^2$] | 130 [km] | 0.43 [1/km] |
| | 2011 | 52 | 299 [$km^2$] | 130 [km] | 0.43 [1/km] |
| | 2012 | 54 | 299 [$km^2$] | 130 [km] | 0.43 [1/km] |
| | 2013 | 55 | 256 [$km^2$] | 77 [km] | 0.30 [1/km] |
| | 2014 | 53 | 257 [$km^2$] | 100 [km] | 0.41 [1/km] |
| | 2015 | 58 | 257 [$km^2$] | 100 [km] | 0.41 [1/km] |
| | 2016 | 53 | 257 [$km^2$] | 130 [km] | 0.51 [1/km] |
| | 2017 | 46 | 256 [$km^2$] | 110 [km] | 0.41 [1/km] |
| | 2018 | 55 | 256 [$km^2$] | 130 [km] | 0.52 [1/km] |
| | Total | 662 | 3500 [$km^2$] | 1500 [km] | 0.42 [1/km] |
| Yellowtail | 2007 | 29 | 214 [$km^2$] | 100 [km] | 0.49 [1/km] |
| | 2008 | 29 | 214 [$km^2$] | 100 [km] | 0.49 [1/km] |
| | 2009 | 29 | 128 [$km^2$] | 65 [km] | 0.51 [1/km] |
| | 2010 | 29 | 171 [$km^2$] | 91 [km] | 0.53 [1/km] |
| | 2011 | 28 | 128 [$km^2$] | 65 [km] | 0.51 [1/km] |
| | 2012 | 29 | 128 [$km^2$] | 65 [km] | 0.51 [1/km] |
| | 2013 | 29 | 300 [$km^2$] | 140 [km] | 0.48 [1/km] |
| | 2014 | 29 | 300 [$km^2$] | 150 [km] | 0.48 [1/km] |
| | 2015 | 29 | 300 [$km^2$] | 150 [km] | 0.48 [1/km] |
| | 2016 | 29 | 300 [$km^2$] | 150 [km] | 0.48 [1/km] |
| | 2017 | 29 | 300 [$km^2$] | 150 [km] | 0.48 [1/km] |
| | 2018 | 29 | 300 [$km^2$] | 150 [km] | 0.48 [1/km] |
| | Total | 347 | 2780 [$km^2$] | 1400 [km] | 0.49 [1/km] |

Since 1996, the scallop fishery has only been permitted to land Atlantic sea scallop and monkfish (*Lophius americanus*), while incidental catches of other species are required to be discarded. Discards from the scallop fishery on Georges Bank are estimated from at-sea observer coverage of the fishery. Observer coverage for the scallop fishery on Georges Bank consists of two observed trips per month, as required by DFO.

License conditions for the scallop fishery require the use of Vessel Monitoring Systems (VMS) at a polling rate of 1 hour. VMS data were available from 2000 onward, and were obtained from the DFO database in August of 2019. VMS data included all vessels in the scallop fleet, identified and verified by vessel name and vessel identification numbers. The records before 2005 were not complete and some VMS data were supplemented with locally maintained records; pre 2005 up to 40% of the VMS records had to be supplemented with these locally maintained records. Since 2005, over 99% of the VMS records were available. Raw VMS data were filtered for duplicates and to remove pings that occurred less than 48 minutes since the previous ping. Great circle distances were calculated based on differences between subsequent locations and were used, along with the time difference, to estimate vessel speed (knots). All VMS records within the Georges Bank defined area were included along with the VMS records on all other offshore banks fished by the scallop fishery, defined by the extent of

the DFO scallop survey strata (Fig 1). Discussions with the scallop industry suggested that fishing occurs at speeds less than 5 knots, thus only VMS records with speeds $< 5$ knots have been retained to identify fishing activity.

## Statistical analyses

For all of the spatial analyses, Georges Bank was subdivided into a grid that aligned with the closure cells and all analyses were performed using data aggregated at the scale of the closure cell grids. All analyses were performed in R. The sf package was used for spatial analyses, the mgcv package for the GAM models, and the figures were developed using ggplot2 [50–53].

**Fishery effort.** The pre-closure era (2000-2005) was treated separately from the closure era (2006-2018). In the pre-closure era, the analyses of the cod and yellowtail closures incorporated all cells that were subsequently included in the cod or yellowtail closures. In the closure era only those cod or yellowtail cells that were closed in a particular year were used; the effort estimates for inside the closure cells are based on the portion of the year in which the closure was not active.

VMS data were used to determine the location of fishing activity as it provides a frequent (hourly) unbiased indicator of the location of each fishing vessel. A similar procedure was followed to obtain the total effort estimates by era for each cell. The VMS data ($< 5$ knots) was used to determine the total time spent fishing within a cell ($E_c$; in hours) based on the time difference between successive VMS polls ($P_c$) associated with each VMS record ($v$) in a given cell in each year.

$$E_{c,y} = \sum_{v=1}^{n} P_{c,y} \tag{1}$$

To get the VMS effort by era in each cell ($E_{c,era}$) the average effort in the cell was taken for all the years associated with each era.

$$E_{c,era} = \begin{cases} \bar{E}_{c,pre} & \text{when } 2000 \leq y \leq 2005 \\ \bar{E}_{c,closure} & \text{when } 2006 \leq y \leq 2018 \end{cases} \tag{2}$$

The VMS effort in each cell was classified into 4 categories based on the proportion of total annual effort. Cells in which the proportion of total effort was in the lowest quartile ($<25\%$) were classified in the *Low* category, cells in the second quartile were classified as *Below median*, cells in the third quartile were classified as *Above median*, and the *High* category was for the cells in which the proportion of effort was in the top 25%.

**Scallop productivity.** Since 1981 DFO has conducted a summer (August) scallop survey on the Canadian portion of Georges Bank to estimate and monitor scallop biomass. The survey is a stratified random design and currently uses the historic survey index (1981-2009) to define the strata. The survey currently samples at least 230 stations each year. All scallop caught on the survey are counted and individual shell heights recorded, a subset of scallop are further processed to measure their meat weight. The relationship between meat weight and shell height is then used to estimate the total survey biomass for each station (in terms of meat weight). For more details on these calculations and the scallop survey see [54].

Due to the low mobility of adult scallop, biomass can be used as a measure of productivity within an area. Here, biomass from the scallop survey (for scallop with a shell height $\geq 95$ mm) was used to determine the scallop productivity of each cell ($c$) on the bank. Within each cell, the average biomass per tow in a given year (y), $BPT_{c,y}$, was calculated. This was then

multiplied by the towable area $A_c$ of the cell to obtain the total biomass in a cell for each year.

$$B_{c,y} = BPT_{c,y} \times A_c \qquad (3)$$

To get the biomass per cell by era ($B_{c,era}$) the average biomass in the cell was taken for all the years with survey data in a cell during each era.

$$B_{c,era} = \begin{cases} \bar{B}_{c,pre} & \text{when } 2000 \leq y \leq 2005 \\ \bar{B}_{c,closure} & \text{when } 2006 \leq y \leq 2018 \end{cases} \qquad (4)$$

Scallop productivity was classified into 3 categories based on biomass. Cells that had a below average proportion of the total annual scallop biomass were categorized as *Low* because the fishery does not regularly target cells with below average biomass. Cells corresponding to the most productive 25% of the bank during the study period were identified as *High* productivity cells. The remainder of the bank was classified as *Medium* scallop productivity.

**Relationship between fishery effort and scallop productivity.** The VMS fishing effort data were compared to scallop biomass to describe the fishery in the two defined eras (pre-closure versus closure). The relationship between monthly VMS effort and scallop productivity (in terms of biomass) was compared spatially and between eras to determine if there was evidence that the closures influenced the fishing patterns of the scallop fleet on Georges Bank.

The fishery can also direct effort to other banks (Fig 1). To determine if the closures resulted in effort displacement from Georges Bank to other banks the monthly VMS effort on Georges Bank was compared to the total monthly VMS effort on the other banks in the pre-closure and closure eras.

To determine the extent to which the closures led to small scale displacement of effort on Georges Bank itself, an analysis was performed to compare the monthly VMS *Effort* ($hours \times km^{-2} \times day^{-1}$) inside the closure cells in the month before, during, and the month after the closure (*timing*) with the effort on the rest of the bank (*loc*). The model was a *Gamma* GLM, where the $\phi$ parameter controls the shape of the *Gamma* distribution, while $\mu_i$ is the expected value of *Effort$_i$*. In some years there was no effort inside the closure cells. Given that a *Gamma* GLM requires strictly positive values, the years with no effort were set to a small positive value ($\approx 10\%$ of the minimum observed effort in a cell).

$$Effort_i \sim Gamma(\mu_i, \phi)$$

$$E(Effort_i) = \mu_i \quad and \quad var(Effort_i) = \frac{\mu_i^2}{\phi} \qquad (5)$$

$$log(\mu_i) = loc_i \times timing_i$$

**Fishery landings.** For this analysis the logbook data were used to calculate the monthly scallop landings on Georges Bank from 2000–2018. A *beta* GLM with a logit link was used to model how the cumulative monthly proportion of the total annual scallop landings (*PTL*) varied by *month* between each *era*. The total fishery landings were used instead of the TAC since landings have been within 1.5% of the TAC during the study period and better represent the total removals from the fishery each year. The monthly cumulative landings and the total annual landings for each year were calculated from the logbook data. The *beta* distribution cannot include zeros or ones so the data were transformed following [55]. The $\psi$ parameter

controls the shape of the *beta* distribution;

$$PTL_i \sim beta(\mu_i, \psi)$$

$$E(PTL_i) = \mu_i \quad and \quad var(PTL_i) = \frac{\mu_i \times (1 - \mu_i)}{\psi + 1} \tag{6}$$

$$logit(PTL_i) = month_i \times era_i$$

**Discards.** The observer program commenced in September 2004 and there was no observer coverage in 2005 during several months (including February and June). As a result, there was insufficient data from the observer program on Georges Bank during the pre-closure era to quantitatively compare how discards from the scallop fishery have been impacted by the implementation of the closures. Further, the coverage and resolution of the observer data preclude their use in identifying fine-scale spatial discard patterns on Georges Bank (two trips per month); however, they can provide insight into the seasonal discard patterns of the scallop fishery. This analysis used the observer data from 2007-2018.

Estimates of discarded weights (*kg*) for Atlantic cod and yellowtail flounder from each observed scallop trip (*o*) were retrieved from the observer database, verified for accuracy, and prorated to account for unobserved portions of observed trips. Fishing effort for each observed trip was calculated using the tow duration and gear width from the logbook data ($Eff_o$, with units of *hour* × *metres* (*hm*)). For each month (*m*) in year *y* the discard rate ($DR_m$ with units of $kg \times hm^{-1}$) was calculated for each species as the sum of the observed trip discards ($D_o$) in month *m* divided by the sum of the observed trip effort ($Eff_o$) in that month:

$$DR_{m,y} = \frac{\sum_{o=1}^{n} D_o}{\sum_{o=1}^{n} Eff_o} \tag{7}$$

From the Georges Bank scallop fishery logbook data, monthly fishing effort ($Eff_m$) was calculated. Monthly discard rates ($DR_m$) were then multiplied by the monthly effort from all fishing trips to estimate the total discards from the fishery each month ($D_m$ with units of metric tonnes, mt):

$$D_{m,y} = DR_{m,y} \times Eff_{m,y} \times 0.001 \tag{8}$$

For months with no observed trips, the previous month's discard rate was assumed. Total annual discards ($D_t$, with units of mt) were calculated as the sum of the monthly discards ($D_m$) in year *y*:

$$D_{t,y} = \sum_{m=1}^{12} D_{m,y} \tag{9}$$

The monthly proportion of total annual discards ($PD_m$) was calculated for each month in year $y$ as:

$$PD_{m,y} = \frac{D_{m,y}}{D_{t,y}} \tag{10}$$

A monthly discard rate anomaly ($DRA_m$) was calculated as the discard rate in each month of year $y$ divided by the median discard rate in that year ($\widetilde{DR}$):

$$DRA_{m,y} = \frac{DR_{m,y}}{\widetilde{DR}_y} \tag{11}$$

The observer data analysis used a generalized additive model (GAM) with a thin plate regression spline smoother to estimate the seasonal patterns of monthly discards ($D_m$), monthly proportion of total discards ($PD_m$), and the monthly discard rate anomaly ($DRA_m$) for both cod and yellowtail. The $D_m$ and $DRA_m$ analyses used a *Gamma* GAM with a log link.

$$D_{m_i} \sim Gamma(\mu_i, \phi) \quad and \quad DRA_{m_i} \sim Gamma(\mu_i, \phi)$$

$$E(D_{m_i}) = \mu_i \quad and \quad E(DRA_{m_i}) = \mu_i$$

$$var(D_{m_i}) = \frac{\mu_i^2}{\phi} \quad and \quad var(DRA_{m_i}) = \frac{\mu_i^2}{\phi} \tag{12}$$

$$log(\mu_i) = s(month_i)$$

$PD_m$ was modelled using a *beta* GAM with a logit link. These data were transformed following [55].

$$PD_{m_i} \sim beta(\mu, \psi)$$

$$E(PD_{m_i}) = \mu_i \quad and \quad var(PD_{m_i}) = \frac{\mu_i \times (1 - \mu_i)}{\psi + 1} \tag{13}$$

$$logit(\mu_i) = s(month_i)$$

## Results

### Impact of the closures on the scallop fishery

The distribution of effort in the scallop fishery differed between eras (Fig 2a and 2b). In the pre-closure era 47% (28 cells) of the southern area had above average effort. This declined to 7.8% (5 cells) of the area in the closure era. Meanwhile, the area fished in the north increased by approximately 13%, while the percentage of the area experiencing above average effort declined slightly from 61% (41 cells) to 57% (43 cells; note the total area of high scallop productivity increased by 2 cells) between the eras. In addition, the total effort in the south declined from 17% of the total annual effort in the pre-closure era, to 8.5% in the closure era.

The changes in the spatial fishery patterns were largely driven by a shift in the distribution of scallop productivity between the northern and southern portion of Georges Bank (Latitude 41.85˚; Fig 2c and 2d). In the pre-closure era 14% (8 cells) of the southern half of the bank was

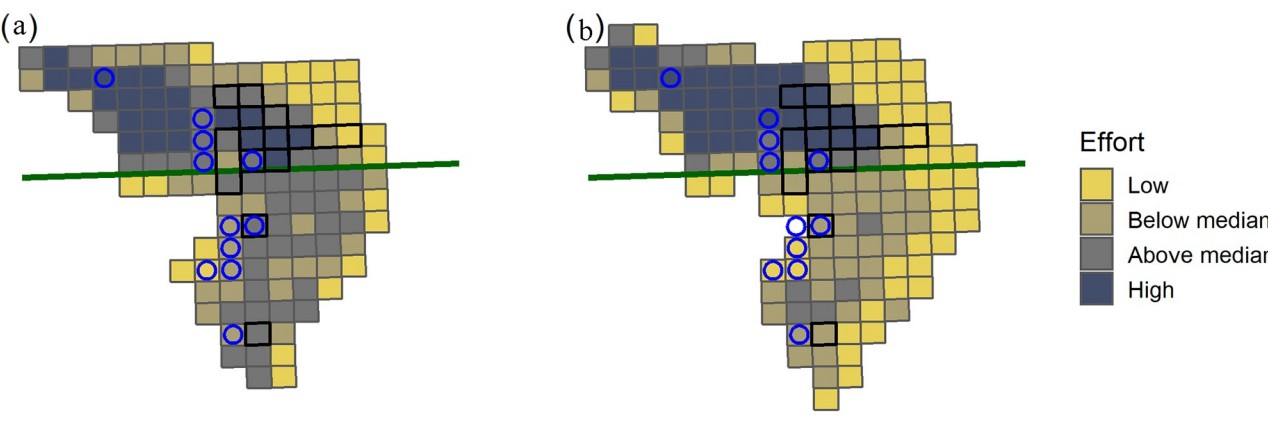

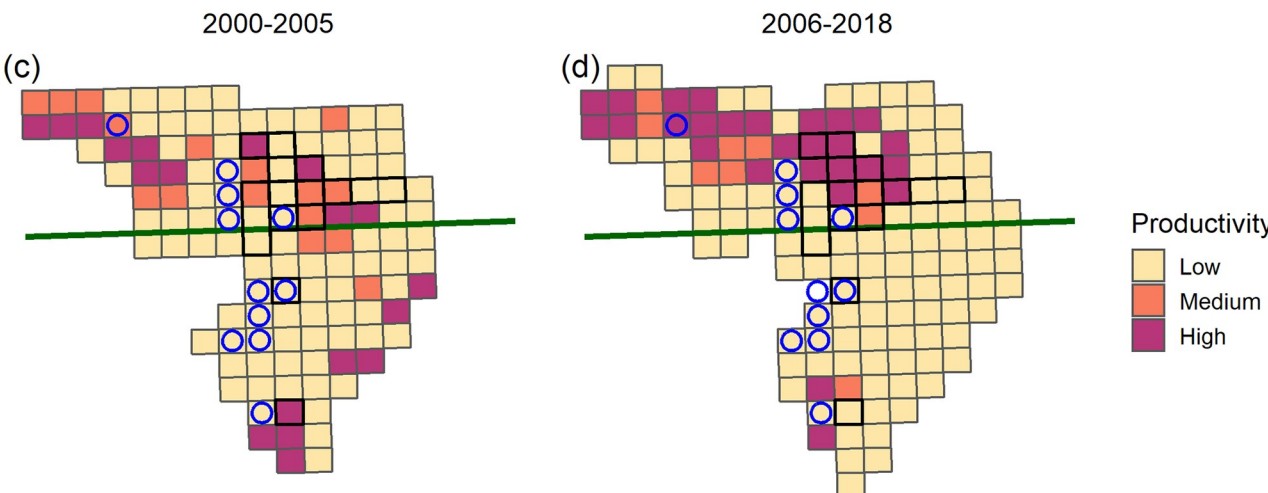

**Fig 2. Spatial distribution on Georges Bank of the average scallop productivity (a,b) and average scallop fishing effort (c,d) in pre-closure (2000-2005) and closure (2006-2018) eras.** The bank was divided into cells that are the same size and overlap with the yellowtail closure cells (indicated by blue circles) and the cod closure cells (indicated by cells outlined by thicker black line). The green line represents the north-south division of the fishing area on Georges Bank.

classified as high scallop productivity. This declined to just 3.1% (2 cells) of the southern area in the closure era. Conversely, in the north the high scallop productivity region increased from 16% (11 cells) of the area in the pre-closure era to 33% (25 cells) in the closure era. The proportion of the total scallop biomass on Georges Bank located in the north increased from 64% to 78% between the eras. The low scallop productivity area was consistent between eras accounting for 60% of the total bank area but only 27% of the total scallop biomass.

In both the pre-closure (2000-2005) and closure (2006-2018) eras the high scallop productivity regions of Georges Bank (Fig 2c and 2d) also experienced elevated fishing effort (Fig 2a and 2b). Overall, the fishery has reduced effort in the lower scallop productivity areas and shifted to the high scallop productivity areas which are now located primarily in the northern portion of the bank (Figs 2 and 3).

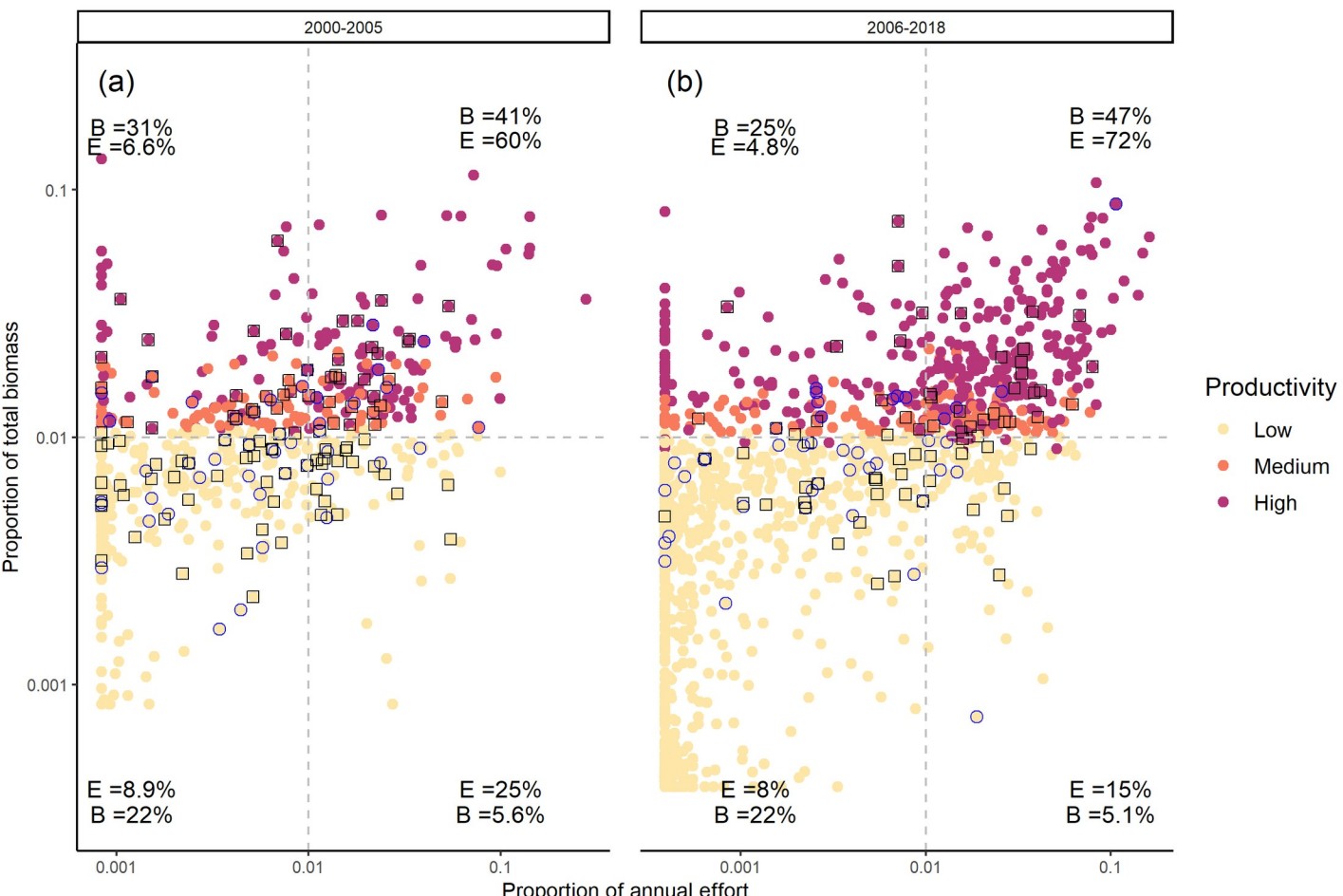

**Fig 3. The relationship between the proportion of total annual biomass and the proportion of total annual effort in the (a) pre-closure (2000-2005) and (b) closure (2006-2018) eras.** Each point represents one of the cells that the bank was divided into in each year. The cells are the same size and overlap with the yellowtail closure cells (indicated by blue circles) and the cod closure cells (indicated by black squares). The colour of each point represents the productivity classification for that cell in a given year. The vertical dashed line represents the mean proportion of total annual effort in the cells, while the horizontal dashed line is the mean proportion of total annual biomass in the cells. Percentages represent the percentage of Biomass (B) and Effort (E) in each quadrant during the respective eras.

In the closure era, the cells that were temporarily closed to protect cod spawning aggregations (Figs 2 and 3), have covered an average of 5.4% of the total scallop fishing area. For the cod closure, 30% of cells were high scallop productivity areas and the effort inside these cells has generally been at or above the bank average with 8.3% of the total annual effort occurring in the cells when they are open. The location of the cod closure cells are predominately in the northern, more productive, region of the bank; on average 82% of the area covered by the cod closures is in this region (Fig 4).

The cells that were temporarily closed to protect yellowtail spawning aggregations, (Figs 2 and 3) have covered an average of 3.4% of the total area fished on Georges Bank each year. Only 17% of the yellowtail closure cells were high scallop productivity areas and the effort inside the yellowtail closure cells has generally been below average for the bank with 2.6% of the total annual VMS effort occurring inside the cells when they are open (Fig 2). The effort for the yellowtail closures is highly influenced by the 2010 closures which was the only year in which an isolated cell in the northwest portion of the bank was closed (Figs 1 and 2). Excluding 2010, effort within the yellowtail closures declined to just 1.7% of the total annual effort. The

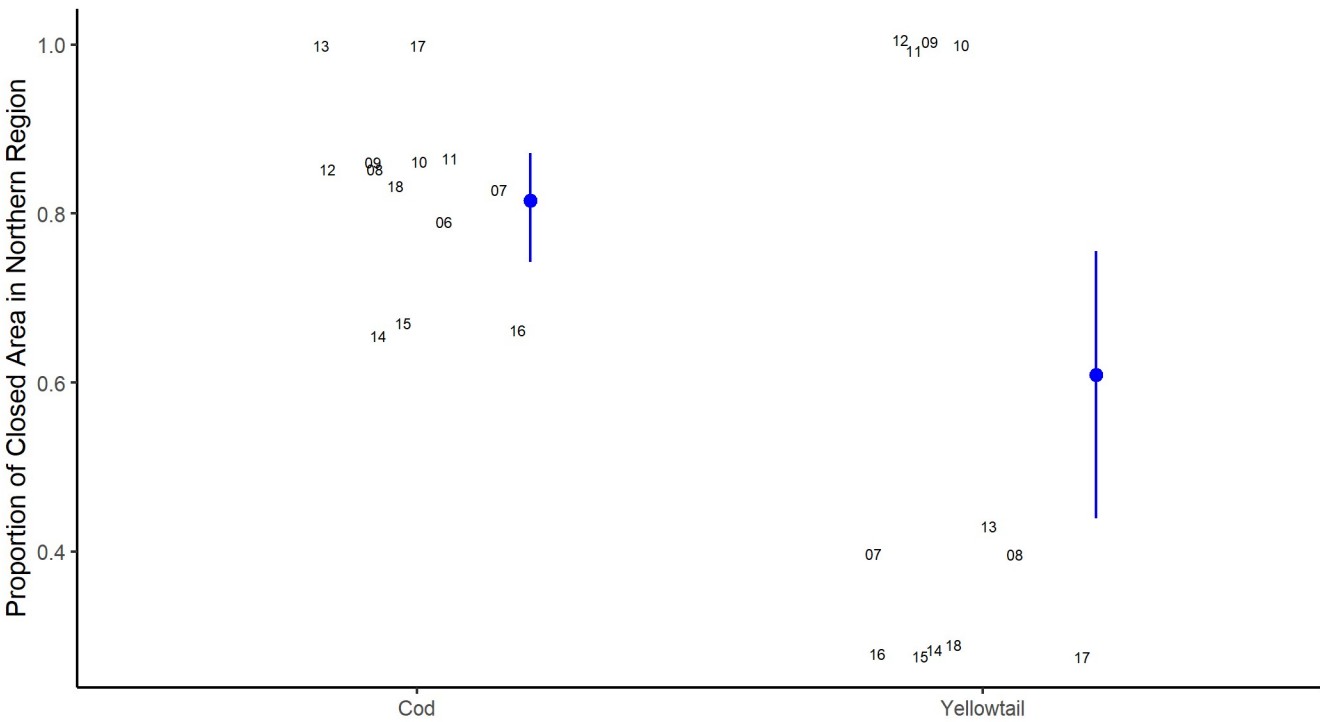

**Fig 4. Annual proportion of the closed cells that were located in the northern half of Georges Bank for the cod and yellowtail closures.** Grey numbers represent the proportion in a given year, the blue point is modelled mean value and error bars represent the 95% confidence intervals. Note the points are jittered slightly to avoid over-plotting and the y-axis does not extend to 0.

location of the yellowtail closure cells were more variable than observed for the cod closure cells (Fig 4). On average 61% of the yellowtail closure cells have been in the north, but this has varied over time, with all of the yellowtail closure cells in the north from 2009 to 2012; whereas it has been <50% in all other years.

In the closure era, the average fishing effort across the bank in the month before (January) the cod closure was active was not significantly different than the effort on the bank during the closure (Fig 5). In the month following the re-opening of the closure (April) effort was elevated relative to the previous periods both inside the closure cells and across the bank, although this difference was statistically marginal given the variability in effort. In the month after the closure, effort was higher in the closure cells than across the bank, but due to the large inter-annual variability of effort inside the closure cells this difference was not significant (Fig 5).

For the yellowtail cells in the closure era, the effort in the month before (May) the closure was active was significantly higher across the bank than inside the closure cells (Fig 5). The effort on the bank when the yellowtail closure was active (June) was not significantly different from either the month before or the month after (July). Effort in July was typically higher on the bank than inside the closure, but due to the large inter-annual variability of effort inside the closure this difference was marginally significant (Fig 5).

For both closures compliance was high, but there were some occasions when vessels entered an active closure cell (Fig 5). When the cod closure was active 99.7% of all VMS effort occurred outside the cod closure cells, while when the yellowtail closure was active 99.8% of the VMS effort occurred outside the yellowtail closure cells. A review of logbook data confirmed that there has been, on occasion, extremely limited fishing within an active closure.

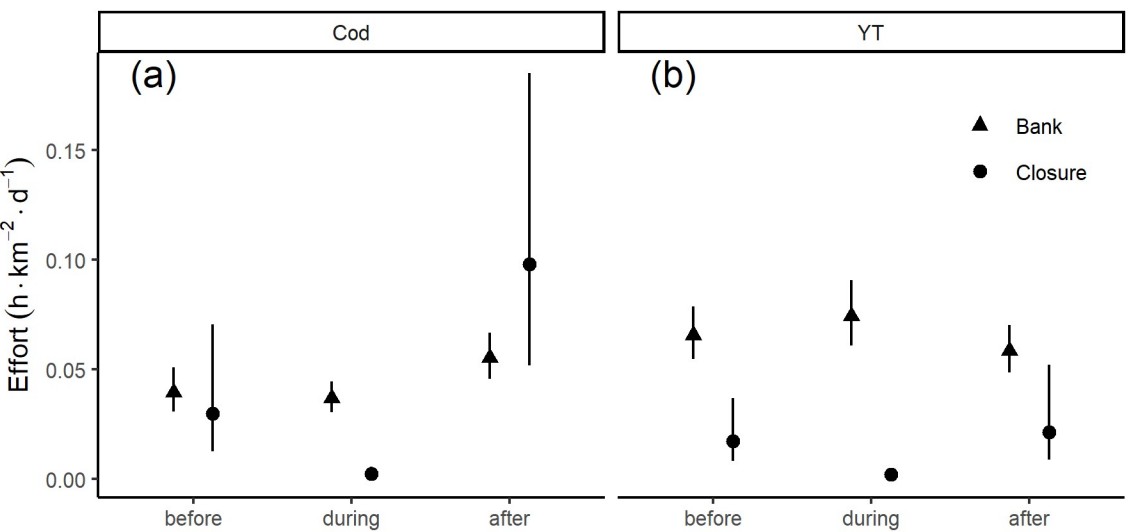

**Fig 5. The scallop fishery effort ($hours \times km^{-2} \times day^{-1}$) in the month before the closure, during, and after the closure; a) cod closures, b) yellowtail closures.** Triangles represent the bank-wide effort while the circles represent the effort inside the closures, error bars represent the 95% confidence intervals.

The majority of fishing activity occurred on Georges Bank, comprising an average of 72% of the total fishery monthly effort across all banks (Fig 6). Within Georges Bank, there was a strong seasonal cycle in effort in both eras; effort peaked during the summer months and was lowest in winter months. During the closure era, the seasonal pattern on Georges Bank shifted, a higher proportion of fishing occurred between April and August than in the pre-closure era. There was a steady decline in effort on Georges Bank from September until December in this era (Fig 6).

Overall annual effort for the offshore scallop fishery has declined in the closure era; however, the relative percentage of effort on Georges Bank increased slightly from 70% to 74% in the closure era. On the other scallop banks in the pre-closure era effort peaked in the summer months, whereas in the closure era the fishery effort was relatively uniform until October after which it declined (Fig 6). The cumulative landings on Georges Bank indicated that 90% of the total annual landings were reached almost a month earlier in the closure era (9.1 months) than during the pre-closure era (10 months; Fig 7).

## Impact of the closures on groundfish discards

There was an order of magnitude decline in the annual total discards for both species during the closure era. The cod prorated discard estimate declined from 100 tonnes in 2007 to 7.9 tonnes in 2018 and yellowtail declined from 92 tonnes to 3.4 tonnes in this same period. Monthly discard estimates of both Atlantic cod and yellowtail flounder on Georges Bank are available for most months since the first year both closures were in place (2007). These estimates indicate that discards of both species followed a seasonal pattern and discards tended to be above average in the month immediately before, during, and the month immediately after the closures were active (Fig 8).

The prorated monthly discard estimate of cod peaked between February and May, with a maximum discard estimate of 4 (CI = 2.6-6.2) tonnes in April (Fig 8a). The proportion of the

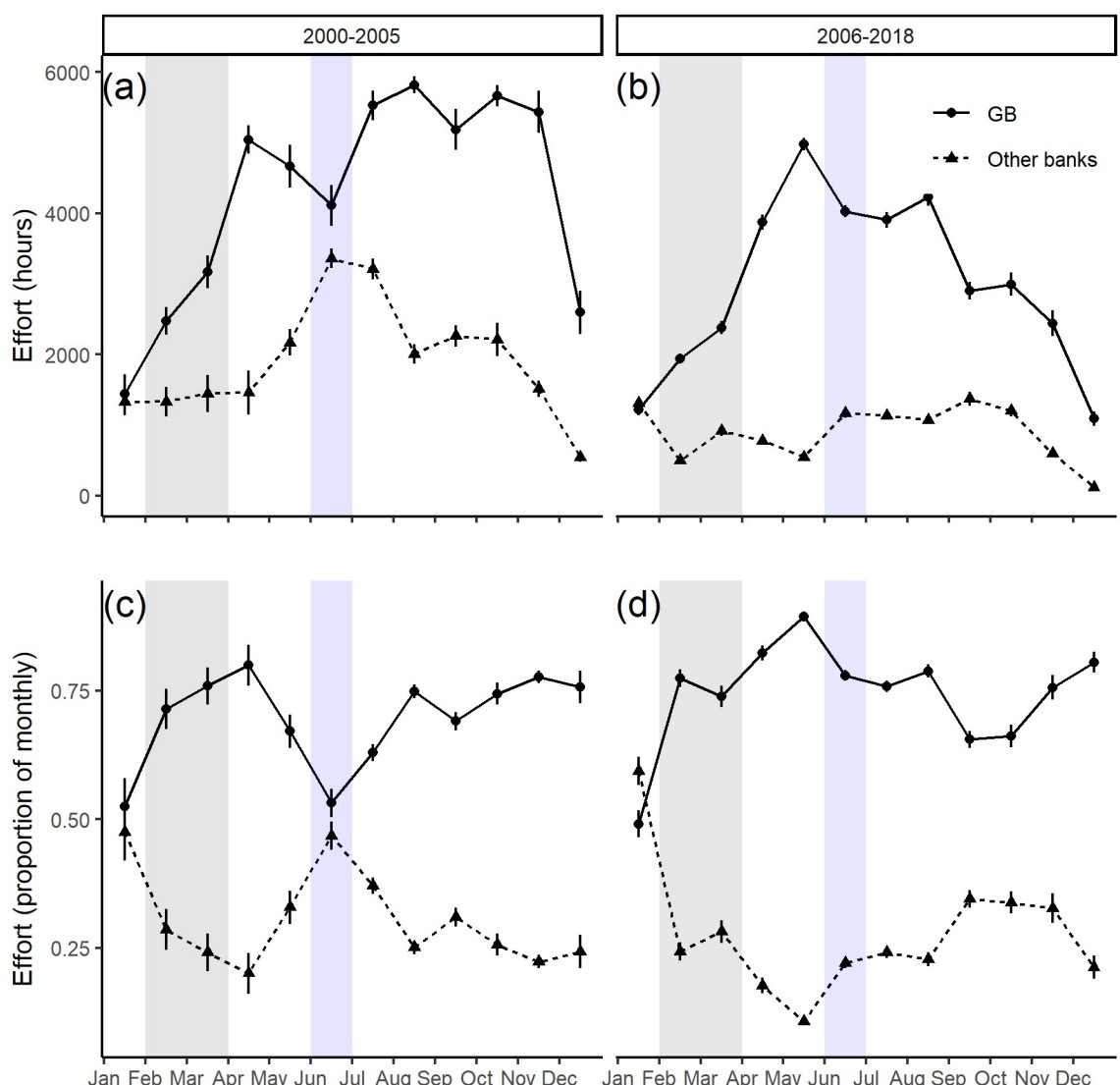

**Fig 6. Monthly scallop fishery VMS effort by bank (GB represents Georges Bank) across years before and after closures were introduced.** Absolute VMS effort (a,b) and proportion of total monthly VMS effort (c,d). The error-bars represent 1 standard deviation from the mean and are intended to indicate the relative annual variability by month in each era. The shaded regions represent the cod (grey) and yellowtail (blue) closure months.

total cod discarded exhibited similar behavior, approximately 48% of the discards occurred between February and May, peaking at 13 (CI = 11-16)% in April (Fig 8c). The discard rate, which is the most direct measure of the frequency of interaction between the fishery and cod, exhibited a slightly different pattern than the discard estimates. The discard rate of cod was highest between January and March and was 2.6 times greater, on average, than the median monthly discard rate. The discard rate was generally above the median from January to May (Fig 8e).

The prorated total discard estimate of yellowtail peaked in April through June, with a maximum discard estimate of 13 (CI = 8-23) tonnes occurring in May (Fig 8b); this result was

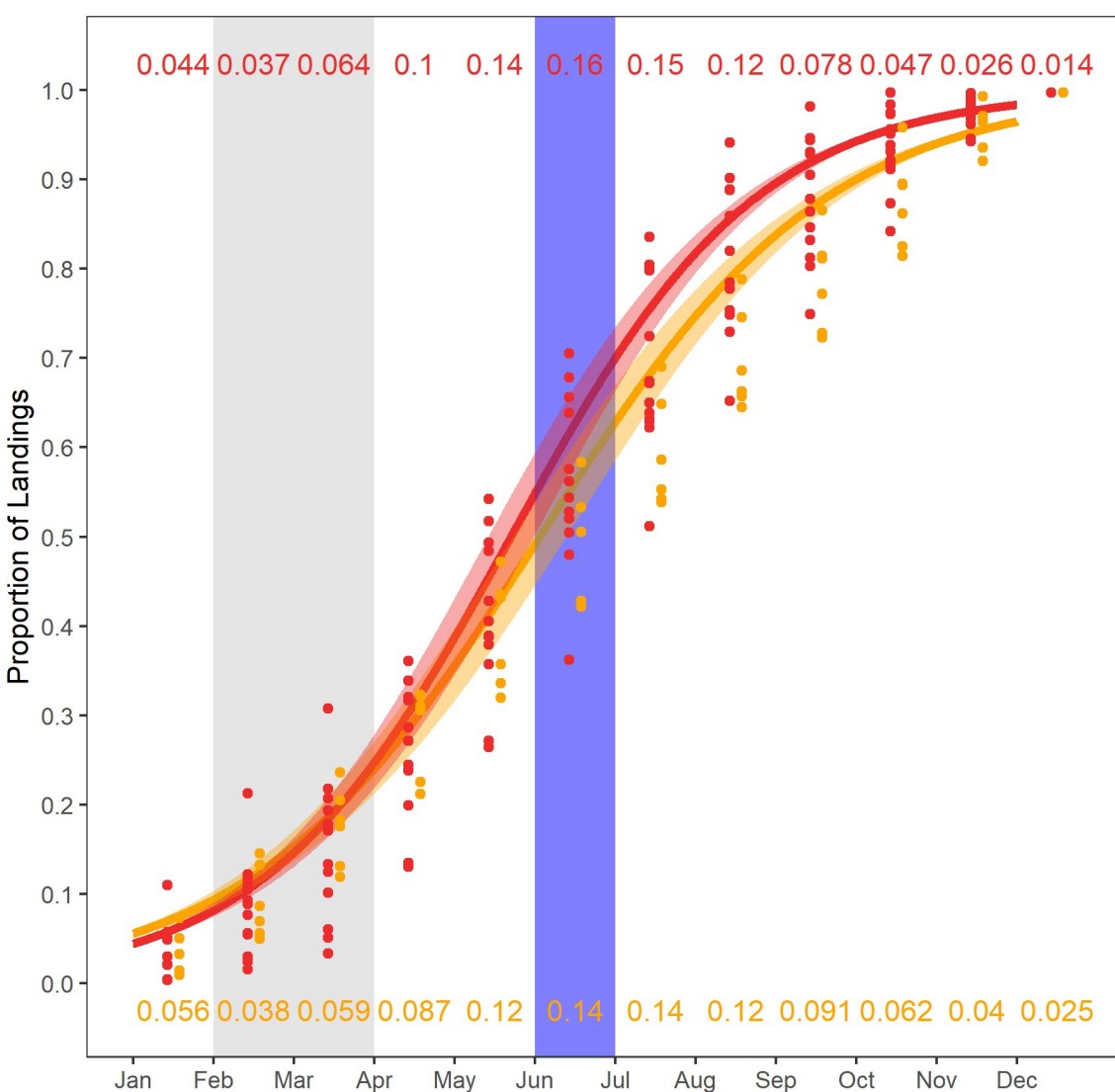

**Fig 7. Cumulative proportion of total scallop landings caught by month in the pre-closure (2000-2005) and closure (2006-2018) eras on Georges Bank.** The pre-closure era is in orange, closure era in red and the shaded region represents the 95% confidence interval for each era. The shaded regions represent the cod (grey) and yellowtail (blue) closure months. The red values along the top of the figure indicate the average proportion of landings in a given month during the closure era, the orange values along the bottom are for the pre-closure era.

highly influenced by the discards observed between 2007 and 2010. The proportion of the total annual yellowtail discards also peaked in April through June with approximately 61% of the total annual discards removed in these months; the peak occurred in May and an average of 25 (CI = 20-30)% of the removals occurred in this month (Fig 8d). Similarly, the discard rate was elevated in these same months, the discard rate in May is 4 (CI = 3-5.4) times the median discard rate (Fig 8f). The combination of elevated discard rates and fishery effort during these months resulted in a very strong seasonal signal in the discards of yellowtail flounder, with peak discards occurring when the yellowtail closure was active.

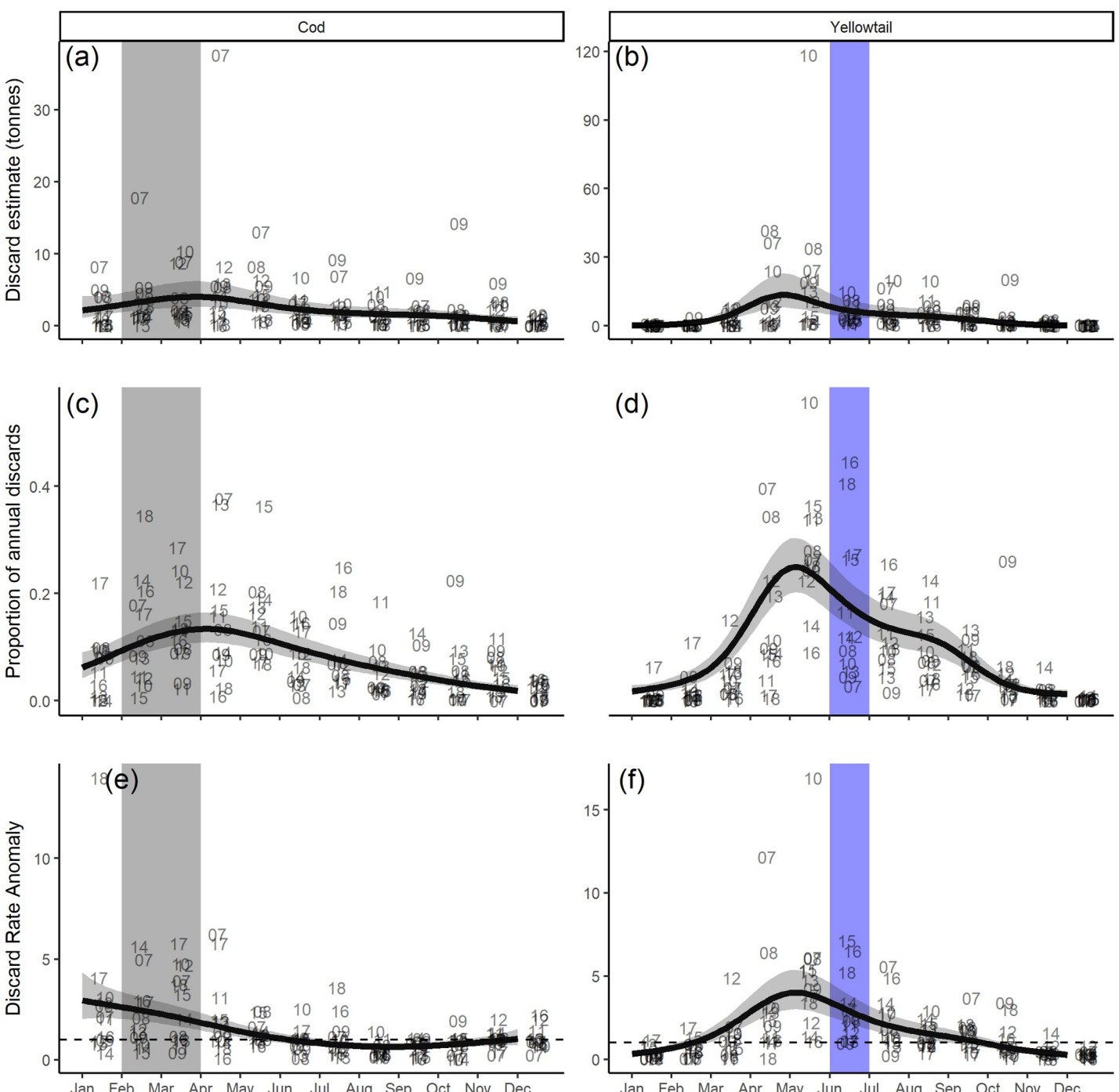

**Fig 8. Observed monthly a) cod prorated total discards, b) yellowtail prorated total discards c) cod proportional discards, d) yellowtail proportional discards, e) cod discard rate anomalies, and f) yellowtail discard rate anomalies on Georges Bank.** Data used were from 2007-2018. Proportions in c) and d) are of the total annual discards from the scallop fishery on Georges Bank in a given year. Anomalies in e) and f) are calculated as a proportion of the annual median discard rate (e.g. a value of 2.5 represents a discard rate that is 2.5 times larger than the median monthly discard rate for that species in a specific year). The numbers represent the year of observation. The model trend is represented with the black line and the 95% confidence interval is the shaded area around this line. The highlighted months represent the month in which the respective closures were active.

## Discussion

The primary goal of this paper was to evaluate the effectiveness of time-area closures in achieving their conservation objectives while assessing the socio-economic impact of the closures on the Canadian Georges Bank scallop fishery. The conservation objective for the closures was to protect spawning aggregations of Atlantic cod and yellowtail flounder during peak spawning periods from the mobile bottom gear deployed by the scallop fishery. We found that time-area closures had minimal impact on the scallop fishery, however these results suggest that the conservation objectives of the closures are not being fully realized. Our methods demonstrate an evidence-based approach which could benefit the design, implementation, and evaluation of SFM strategies and could be adapted for the development of a broader evaluation framework.

### Impact of the closures on the scallop fishery

The scallop fishery effort closely followed the spatial scallop productivity patterns as has been observed in other scallop fisheries [56, 57]. The scallop fishery has increasingly focused effort in areas of high scallop biomass since the pre-closure era. The cod closure cells are generally located in a moderately productive region of the bank for scallop and the effort in these cells is above the bank average throughout most of the year. However, the scallop fishery effort on Georges Bank when the cod closures were active has historically been relatively low; the scallop fleet fishing effort on Georges Bank typically increases in April concomitant with an increase in scallop meat condition [58]. In the pre-closure era 14.7% of the TAC was harvested during February and March when the cod closure was active. This increased slightly to 16.6% during the same months in the closure era. While the cod closure cells were often in a relatively productive area of the bank for scallop, this closure was active before the peak of the scallop fishery and thus had little impact on the behavior of the fishery over the course of the fishing season. Conversely, the yellowtail closure cells were generally in a less productive area of the bank for scallop, but were active when fishing effort was relatively high. In the pre-closure era on average 13.7% of the TAC was removed in June. This increased slightly to 15.2% in June during the closure era. Despite the high scallop fishing effort when the yellowtail closures were in place, the location of the yellowtail closure cells in a region of relatively low scallop productivity on Georges Bank resulted in a low impact on the scallop fishery. Although the causes differ, this evidence suggests that both closures have not had a significant impact on the scallop fishery.

A lack of direct monitoring data poses challenges to evaluating the effectiveness of time-area closures since monitoring data is needed to evaluate both socio-economic and conservation objectives. A lack of evidence of socio-economic impact on its own does not necessarily indicate that a closure is (in)effective; however, a lack of direct monitoring data poses challenges to evaluate whether a closure is responsible for any observed change [2]. In this case, without the scallop survey data the redistribution of the scallop fishery to the northern portion of the bank may have been attributed to the closures rather than the changes in scallop productivity patterns. More generally, without proper monitoring, coincidental changes in the state of the environment can obfuscate the effect of a closure on a fishery [41]. As demonstrated here, changes may also occur at relatively small spatio-temporal scales, compared to the synoptic scales at which stocks are typically assessed and managed. Monitoring strategies would therefore benefit from accounting for spatio-temporal variability at the appropriate scales and recent statistical advances in spatial modelling should help to facilitate these types of analyses [59, 60].

## Impact of closures on groundfish discards

If the closures were effective in reducing discards an anomalous decline in the discard metrics would be expected during the period in which the closures were active, however, this was not observed. When the closures were active the discard rates remained above average and similar to the months before and after they were active. Similarly, an above average proportion of the total annual discards were landed in the months the closures were active. In recent years, the proportion of total annual discards for both species has been elevated when the closures were active; greater than 30% and greater than 40% of the total annual discards were observed when the closures were active in 2016-2018 and in 2016 and 2018, for Atlantic cod and yellowtail respectively. Although it is possible that discards while the closures were active may have been higher during the closure era had the closures not been in place, we cannot draw this conclusion due to the lack of baseline observer data from the pre-closure era.

The observed temporal patterns in discard metrics are in agreement with the historically observed spawning patterns for both species. Atlantic cod spawning aggregations are believed to last from November to May with a peak in February and March, while yellowtail flounder spawning is thought to last from April to August with a peak in May and June [48]. However, discard metrics were elevated both before, during, and after the closures were active and the peak discard metrics were generally observed the month before or after the closures were active. These results indicate that the duration of the closures was insufficient to provide protection for the duration of the peak spawning season. The weight of evidence from the discard metrics suggests that the efficacy of the current closure design in reducing by-catch is limited and that both the spatial and temporal scale of the closure could be improved. Appropriate spatial scale is one of the most commonly discussed aspects of closure design and the development of closures which are too small to meet SFM objectives is commonplace [27, 61]. For mobile species making seasonal migrations [31, 33] and when a closure is likely to lead to fishing effort near the closure boundaries [34, 35], a thorough analysis of spatial patterns and processes should be a primary consideration during the design of protected areas [9, 33, 36]. An appropriate closure duration is another commonly observed shortcoming of time-area closures [31] and is especially problematic when effort increases in the period after the closure is re-opened [37].

Despite the apparent limited efficacy of the current design of the groundfish closures, during the closure era the discards of both Atlantic cod and yellowtail flounder in the scallop fishery have declined by an order of magnitude. In contrast to the above evidence, this would suggest that these closures were effective. However, the effort in the scallop fishery has been in decline over this time, with the annual effort in recent years (2014-2018) approximately 50% lower than the effort in the pre-closure era. In addition, since the introduction of these closures the biomass of Atlantic cod in the region has been low [42]. The lack of recovery of other cod stocks in the Northwest Atlantic have been linked to elevated rates of natural mortality [62, 63]. Elevated mortality, despite a decline in fishing mortality, has been suggested as a factor inhibiting recovery of Atlantic cod and additional SFM measures may be required to facilitate its rebuilding [64]. Similarly, the biomass of yellowtail flounder on Georges Bank is currently near historical low levels. Despite the yellowtail closures being implemented in 2007, significant declines in yellowtail biomass have continued [43, 65, 66]. This decline followed a rapid recovery in yellowtail biomass in the early 2000's on both the Canadian and U.S. portions of Georges Bank; the recovery was attributed to a decline in fishing mortality and favourable environmental conditions [66]. While there have been a number of hypotheses postulated for the latest decline of this stock, given the decrease in yellowtail discards from the scallop fisheries observed in both Canada and the U.S. [67], it is unlikely that discards from the scallop

fisheries contributed significantly to the decline and the ongoing lack of recovery during this period.

## Other management options

Properly designing a time-area closure requires knowledge of the spatial and temporal dynamics of the resource being protected along with an understanding of the dynamics of the fishery being targeted by the closure [9, 15]. Murray et al. [31] provide a series of conditions under which time-area closures can be effective in reducing by-catch. For time-area closures, such as those presented here, to achieve conservation objectives by-catch patterns must be predictable, both in space and time, and the majority of by-catch should occur in a small, predictable subset of the whole fishing area [31]. Displacement of the fishery to other areas or times of the year with similar (or higher) by-catch rates limits the conservation effectiveness of a closure and can result in unintended negative consequences both to the fishery and the ecosystem [31, 35, 37, 41, 68–70]. Finally, the negative conservation effects of a spatial mismatch between the closure and the resource it is attempting to protect is amplified when closures are isolated in small patches [71]. The evidence here suggests that the cod and yellowtail closures on Georges Bank suffer from many of the deficiencies that have resulted in ineffective time-area closures in other jurisdictions.

To be effective, time-area closures should account for variability in discard risk in both space and time across the fishery's domain [15, 33, 36]. For the scallop fishery on Georges Bank, the fishery and discard patterns suggest that these closures have not significantly impacted this discard risk. Therefore, alternative SFM strategies may prove more effective in reaching conservation objectives without imposing undue socio-economic restrictions on the fishery. Recently, the concept of dynamic ocean management has become increasingly popular as a means to attempt to minimize socio-economic costs while maximizing conservation benefit. This SFM tool accounts for the dynamic nature of marine environments through the incorporation of monitoring and management in near real-time [17, 19]. The underlying premise of dynamic management is that management conducted at the scale of the relevant biological, environmental, and anthropogenic variability will improve conservation benefit while enabling a fishery to operate with minimal additional cost [19, 72]. Increasingly, dynamic ocean management techniques are being developed and implemented to improve SFM of marine resources [40, 67, 73].

On the U.S. side of Georges Bank a dynamic management program was implemented in 2010; this was a formal by-catch monitoring program in which yellowtail flounder by-catch in the U.S. scallop fishery was monitored in near real-time [67]. This program relied on a close collaboration with industry and the development of a fleet communication system so that industry could provide daily by-catch updates; these updates were summarized and maps were provided to the fishery to indicate the risk of encountering yellowtail throughout the region. This collaboration was effective in reducing by-catch of yellowtail flounder largely due to the desire of industry to reduce economic losses associated with closures of highly productive scallop habitat in the previous decade [18, 67]. The program was deemed a success as it reduced both discards and the economic losses due to closures during the early years of its implementation [67].

In the later years of the yellowtail monitoring program in the U.S., the effectiveness declined due to a lack of industry support; this was due to the industry perception that the risk of economic loss had declined. The perceived reduction in risk was likely due to a decline in the likelihood of encountering yellowtail concomitant with the decline in yellowtail flounder biomass on Georges Bank; these declines are consistent with the declines in discards observed

by the scallop fleet in Canada during this same period. Other by-catch mitigation programs using fleet communication have proven successful as long as there is a strong incentive for the fishery to remain engaged in the partnership [74, 75]. The experience in the U.S. scallop fishery is an example of how a dynamic management program requires carefully developed long-term incentives to be successful [76]. The long-term success of a dynamic management strategy requires sufficient planning, consultation, and compliance from stakeholders due to the often intensive nature of the strategy.

In Canada, a similar, yet informal, dynamic management program has also been implemented but without direct management involvement. The scallop fishery has established an informal *voluntary move protocol* in which vessels will move and inform their counterparts if they encounter unusually high groundfish by-catch in a particular area. The primary advantage of this strategy is that it requires no additional resources to implement, but unfortunately, this also means that no data are collected to quantify its effectiveness. While not directly quantifiable, the cumulative effect of, a) the *voluntary move protocol*, b) the significant overall decline in scallop fishery effort, and c) this effort being more concentrated in high scallop productivity zones, are likely to have contributed more to the observed reduction in discards of cod and yellowtail in recent years than the time-area closures.

## Conclusions

These results show how fishery dependent and fishery independent data can be used to assess the impacts of time-area closures on socio-economic and conservation objectives even without the establishment of a dedicated monitoring program. For our case study of the cod and yellowtail closures on Georges Bank the evidence suggests that the spatial and temporal extent of these closures were insufficient to significantly reduce discards for either species during spawning and that neither closure has had a consequential impact on the scallop fishery. While there has been a steady decline in the annual discards from the scallop fishery during the closure era, the drivers of this decline are likely unrelated to the closures themselves. Improvements to the closures design and/or alternative strategies are likely required to further protect the spawning aggregations of these groundfish stocks. The methods demonstrated in this study provide a framework that leverages existing data streams to evaluate the effectiveness of closures relative to SFM objectives.

## Acknowledgments

We thank Amy Glass, Alan Reeves, Dheeraj Busawon, and Irene Andrushchenko for valuable discussions and insights. We also thank the editor and the two reviewers whose thoughtful reviews greatly improved this manuscript.

## Author Contributions

**Conceptualization:** David M. Keith, Jessica A. Sameoto, Christine A. Ward-Paige.

**Data curation:** David M. Keith, Jessica A. Sameoto, Christine A. Ward-Paige.

**Formal analysis:** David M. Keith, Freya M. Keyser, Christine A. Ward-Paige.

**Funding acquisition:** David M. Keith, Jessica A. Sameoto.

**Investigation:** David M. Keith, Jessica A. Sameoto, Freya M. Keyser, Christine A. Ward-Paige.

**Methodology:** David M. Keith, Jessica A. Sameoto, Freya M. Keyser, Christine A. Ward-Paige.

**Project administration:** David M. Keith, Jessica A. Sameoto.

**Resources:** David M. Keith.

**Software:** David M. Keith, Freya M. Keyser.

**Supervision:** David M. Keith, Jessica A. Sameoto.

**Validation:** David M. Keith.

**Visualization:** David M. Keith, Freya M. Keyser.

**Writing – original draft:** David M. Keith, Christine A. Ward-Paige.

**Writing – review & editing:** David M. Keith, Jessica A. Sameoto, Freya M. Keyser, Christine A. Ward-Paige.

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
