## [Decision Letter · Decision Letter 0]

27 May 2020

PONE-D-20-11298

Evaluating socio-economic and conservation impacts of management: A case study of time-area closures on Georges Bank

PLOS ONE

Dear Dr. Keith,

Thank you for submitting your manuscript to PLOS ONE. After careful consideration, we feel that it has merit but does not fully meet PLOS ONE’s publication criteria as it currently stands. Therefore, we invite you to submit a revised version of the manuscript that addresses the points raised during the review process.

In particular, please respond to Reviewer #2’s significant concerns about the validity of conclusions drawn about achieving conservation objectives. Also, in the response to PLOSONE’s Data Sharing question, please be specific about which data can and cannot be shared publicly. The current response is: “Some of the data cannot be shared publicly because of privacy restrictions and collaborative agreements with entities which collected the data.”

We look forward to receiving your revised manuscript.

Kind regards,

Carrie A. Holt

Academic Editor

PLOS ONE

Journal Requirements:

Reviewers' comments:

Reviewer's Responses to Questions

**Comments to the Author**

1. Is the manuscript technically sound, and do the data support the conclusions?

Reviewer #1: Yes

Reviewer #2: Partly

2. Has the statistical analysis been performed appropriately and rigorously? 

Reviewer #1: Yes

Reviewer #2: Yes

3. Have the authors made all data underlying the findings in their manuscript fully available?

Reviewer #1: No

Reviewer #2: No

4. Is the manuscript presented in an intelligible fashion and written in standard English?

Reviewer #1: Yes

Reviewer #2: Yes

5. Review Comments to the Author

Reviewer #1: In their manuscript, the authors describe a very interesting application of fishery dependent and independent data to assess the socio-economic and conservation impacts of time-area closures on Georges Bank. This paper was a pleasure to read and I appreciated the general implication that closures need to be carefully planned and monitored to ensure they are effective. Unfortunately, as the authors point out, closures are sometimes initiated without a plain for monitoring their success. Despite this common reality, the manuscript provides some hope that practitioners can assess the impacts of closures using data from common fisheries monitoring programs. In their case study, several data sources were utilized to assess the impacts of the closures on the fishery (logbooks, catch statistics, VMS, scallop surveys) and the results from several careful analyses show that there are no clear impacts. The conservation implications of the work was also clear, however, the analyses were hampered to a degree by the duration, spatial resolution and coverage of the observer program. For instance, if the series were longer, the authors could have compared bycatch rates in the pre and post closure eras. This limitation precluded the use of a control within the same space. I wonder, however, if the authors considered comparing bycatch rates on the Georges Bank with those from other banks? While this would not control for spatial confounds, it could be an informative comparison. I am also curious whether maturities are ever recorded for bycaught cod or yellowtail as the objective is to minimize bycatch of spawning individuals. Finally, are bycaught species recorded in the logbooks? Apologies for all the questions. I realize that my curiosity may not be helpful here and I suspect the authors have explored these ideas and know that the data are either not available or they are unreliable. I simply raise these questions, just in case, as they may help strengthen their argument that the conservation objectives are not being met.

Data limitations are, of course, no fault of the authors and they only bolster the importance of planning and coordinating monitoring efforts to ensure the objectives of a closure can be evaluated. I reiterate that the authors have done a great job pulling together multiple data streams into one cohesive applied analysis. My only general comment is that the authors should add a little more detail to their model descriptions; in particular, please clarify the unit of analysis. For equations 5 and 6, I could not discern the level of aggregation of the response - does i = vessel, month? For equation 12, I think i = year. Below I provide several specific comments:

# Abstract

Line 24 - Consider removing "and there was no evidence of recovery in either stock" as it implies that there is a link between discards and cod and yellowtail trends (this is ruled out in the discussion).

# Introduction

Line 64 - Consider replacing "SFM objectives" with "both socio-economic and conservation objectives".

Line 96 - I presume bycatch should be added here: "...are small annual [bycatch] quotas allocated...".

# Methods

Lines 131-143 - Consider combining these two paragraph. I initially assumed that the first paragraph was focused on cod and the second on yellowtail but was confused by Lines 139-143.

Line 141 - Spell out the area in km^2^ or add a column to Table 1 for the number of cells.

Line 145 - Consider adding SFA to Figure 1 as some boundaries are mentioned in other figures and throughout the methods.

Lines 163-164 - What is the percent coverage of the observer program, or how many scallop fishing trips are conducted per month?

Line 191 - "...total time spent [fishing] within...".

Line 207 onward - This is perhaps semantics, but when I read productivity I think of growth, recruitment and mortality. Looking at what is presented, I tend to think biomass or density. Perhaps this is a fishy way to think and, for scallops, high productivity is implied by high density.

Lines 264-265 - While it is clear from Figure 8 that there is no clear precipitous decline in bycatch rates during the closures, I wonder if the GAM may be smoothing over a very minor impact. The closed areas are not fished so some, albeit minor, bycatch may be avoided. As the authors note, the crux of the issue is that the size of the closed area is small and the duration too short. Because of this, another covariate is not going to change the story. Nevertheless, consider adding timing as an explanatory variable to get an estimate of the difference in the means. Also consider adding area closed as an explanatory variable as bycatch rates should be lower when more area is closed (then again, there is probably insufficient interannual variability in the area protected to detect a minor signal). Finally, it may be better to use a cyclic cubic regression spline for the smooth for month (i.e. `s(month, bs = "cc")`).

# Results

Line 328 - Might be informative to report some statistics here, such as the parameter estimates of the difference between two groups plus/minus 95% confidence intervals.

Lines 332-336 - Drop the text stating that effort declined slightly because it looks like there is no clear difference in effort on the bank before, during and after the yellowtail closure.

Lines 337-338 - The points in Figure 5 for the during-closure period look very close to zero with high confidence. Perhaps the reference to Figure 5 is not needed here because it is not easy to confirm the assertion given the figure; the percentages from the VMS data serve the purpose.

# Discussion

Lines 408 and 413 - It may be more relatable to simply state the percentages of the landings taken during the cod and yellowtail closures rather than specific quarters or one month.

Lines 516-525 - When was this informal program initiated?

# Conclusions

Lines 528-529 - Suggested tweak of the wording here: "...and conservation impacts of time-area closures even without the establishment of a dedicated monitoring program.".

# Figures

Figure 1 - Add scallop fishing areas if it does not make the map too busy.

Figure 4 - Consider shifting this figure to an appendix.

Figures 6 and 7 - Shaded areas indicating closure windows may be useful in these plots as well.

Reviewer #2: Article Summary

The authors evaluate the impacts of scallop fishery time-area closures to the scallop fishery and to the species of conservation concern, Atlantic cod and yellowtail flounder. Because a monitoring program directed at evaluating these impacts does not exist the authors use scallop fishing effort, location, and discard (of cod and yellowtail) data to evaluate the impacts. They conclude that closures are not impacting the socio-economics of the scallop fishery, and that conservation efforts are not being achieved.

Overall Review

I conclude that the data used by the authors is sufficient and supports the authors conclusions with regards to the impact of closures on the scallop fishery, but that the data is insufficient to evaluate the conservation impacts of the closures on cod and yellowtail stocks in the manner the authors intend. Therefore, I suggest a major revision with regards to the conservation objective.

The discard data, as with any harvest data, is an index or proxy for the population status. Assessing the status of a population from an index without accounting for effort, detection, selectivity, and other data sources provides a limited picture of population status. Additionally, the conservation impact measured using the discard data is not the change in discard from pre-closure to closure eras, but time periods with-in the closure era. This doesn’t give a picture either of the current conservation status of the populations, or the effect of closures on discarding.

I suggest two paths forward for the authors. A) Be clearer that the objective regarding conservation impacts is not to assess the conservation status of cod and yellowtail, but only to assess the impact of scallop fisheries on these populations irrespective to the underlying health of those populations. That is, only draw conclusions on the amount of discarding that is occurring and note methods to further reduce discarding rather than draw conclusions on what the discarding means to the conservation status of species of concern B) Include and evaluate additional data sources regarding the underlying conservation status of cod and yellowtail and evaluate the impacts of discards in this light.

For example, the authors conclude that the closure period is insufficiently long and insufficiently large in area because the proportional discard rate in the time periods before and after the closure are higher than the median rate for the remainder of the year and discarding during the closure period is not reduced relative to the discard rate immediately outside of this closure period. However, these things could be true despite the closure having an impact, and an impact sufficient to sustain the populations of concern. Whether populations are conserved can only be assessed through assessment of the actual status of the populations. Put another way, why would someone care if some discarding is still occurring if the conservation objective is being achieved despite discarding?

Path A for this example would entail retracting the conclusion that the pattern of discards provides information on achievement of conservation objectives. While it does provide some information on the relative amount of discarding, discarding, and discarding by time-period, is an insufficient proxy to measure conservation objectives. Especially the broad unspecified conservation objective discussed here without any associated performance measures. Path B) utilize addition data about the productivity and abundance of cod and yellowtail populations through time to determine how influential the impact of the discards is on the status of the population, and the populations likelihood of achieving conservation performance metrics.

There are also a number of minor changes to figures that would ease interpretation of the results.

Specific Comments

Line 32-34

Awkward phrasing. Maintaining populations above biological reference levels does not maximize socio-economic benefits. Also, biological reference levels, aka biological reference points are a management tool rather than a statement of an objective.

Lines 56-59

Monitoring is not required for effective management, nor are indicators. Without uncertainty in a system, given objectives a management plan could be developed and implemented without the need for monitoring or indicators because the management outcome would be known. Indicators and monitoring are only necessary if there is enough uncertainty in the system that learning can occur from monitoring and that learning will provide a greater benefit than the cost of conducting the monitoring.

Line 66

Components rather than component

Lines 88-90

Presuming the reason for the conservation concern is solely the fishery this would be the case. However, fishing impact is not the only cause for concern for many species of conservation concern, especially those that are primarily by-catch fisheries.

Line 90-91

Suggest rephrasing as “Successful SFM strategies will combine minimal socio-economic impact with maximal conservation impact” because for example, strategies that are worth implementing could be “unacceptable” socio-economically and yet have enough conservation impact to offset them, and vice versa.

Lines 196-199 & 212-217

Why was this data binned into categories, and why were these categories chosen? Unless there was a need for binning, I suggest working with the natural scale data.

Lines 243-245

What about the data was insufficient?

Lines 271-288,306-311, Figure 4

What is the importance of this north-south division? As a reader unfamiliar with SFAs, are northern and southern areas different SFAs. Please clarify, and either way, why would someone care differently about northern and southern areas?

Lines 277-281, Figure 2

Suggest producing an additional pair of images (e and f) showing the proportional effort in an area relative to the proportional productivity (i.e. color scale = proportional effort – proportional productivity)

Lines 291-293

This is the opposite result from the hypothesis that closures are detrimental to scallop fishing. Address this in the discussion. Has the fishery gotten better at finding scallops, did closures have an impact on the fishing behavior that could explain this?

Lines 294-301, Figure 3

If the intended take-away from this figure is that effort is allocated more proportionally to total biomass I suggest replacing the figure with a metric such as Effort deviation = sum across cells(proportional effort – proportional productivity) by era. Could also be calculated for cod and yellowtail areas by era. This would contain more complete information that the % by quadrant.

Line 303-304, 313-314

Reporting % of total productivity in closed cells rather than proportion that are high productivity would provide more information.

Lines 306-311, Figure 4

Either note that the y-axis lower bound is not zero, or alter the bounds on the y-axis to extend to zero.

Lines 323-325, Figure 5

Lower than what? Complete the sentence “…was lower…“ with “than…” For example, “during the pre-closure era”, or “during the closure”. It isn’t clear from the figure, so why is Figure 5 referenced on this line? If during the closure, it doesn’t appear so in Figure 5 where the before and during effort look roughly equal.

Line 338

To me there is no need to reference Figure 5 here, those points are not perceptibly above 0

Line 348

How much has effort throughout the region declined? How has the effort on Georges Bank changed, not as a percentage of regional effort, but absolutely?

Figure 7

If it doesn’t make the figure too difficult to read include the proportion by month as well as the cumulative proportion

Lines 363-364

This implies a major reduction in the impact of fishing on these populations, and therefore potential achievement of the conservation objectives.

That is, unless there was also a 10-fold decline in the cod and yellowtail populations. Another additional way to state the degree of discard would be to report the proportion of the populations lost due to discarding in the scallop fishery. This would provide the missing information on the impact of discards to the populations.

Lines 366-368

It appears it is above average for the three months after the closure for cod and three months before and after for yellowtail.

Figure 8

1. Why do most of the observed data points, way more than 5%, fall outside of the 95% confidence interval?

2. The data in c and e is the same, just transformed. Same for d and f. However, the curves fit in image c and e appear very different. Rather than attempting to fit a spline to these point that assumes some degree of correlation to neighboring months just show the medians for each month.

Lines 398-399

The results show fishing shifted to more productive areas during the closure era, a positive correlation between time-area closures and socio-economic objectives as expressed through the metric of effort allocated to more productive locations.

The results show a 10-fold decrease in discard weight, and a seasonal pattern of discarding weight and rate. Neither provides information on the abundance or productivity of the populations of conservation concern. Therefore, because both the conservation objectives are not specified, and a reduction of discard weight from the scallop fishery presumably is the goal of closures in terms of achieving conservation, the results do not support the conclusion that time-area closures did not achieve conservation objectives.

Line 400-401

The methods here leave out a key step of specify objectives. Without clear objectives performance metrics that measure the status of those objectives will be lacking and monitoring may not measure the parameters of value.

Lines 420-423

Excellent point!

Line 422

Add “in”, to obtain “…coincidental changes in the state…”

Line 427-429

An important note!

Line 433-435

This presumes the amount of discarding would have to be anomalous to be effective. This may not be the case because, 1) effective is not defined, and 2) the discarding that would have resulted without the closure is unknown. Perhaps there would have been an anomalously large amount of discarding during those time periods without the closures. Finally, in the case of yellowtail where fishing seemingly would occur outside of closures anyway, an objective of educating the fishing community about where and when yellowtail spawn to create a conservation mindset, or to practice should the population recover and therefore be subject to additional fishing pressure, could also be an objective and affect the conservation of the population in other ways.

Line 440-443

Enough and adequate are value statements that have not been defined and no information is provided about the spatial distribution of cod or yellowtail. This conclusion is not supported. The only conclusion that can be drawn from the data provided is that the closures do not eliminate by-catch and discarding of cod and yellowtail. However, that amount of reduction in by-catch may be more than enough or exceed what is adequate.

Line 465-466

Declines in yellowtail biomass

Lines 457

“Despite incomplete elimination of discarding during these closures, …”

Line 458-459

Yes.

Line 459-466

How low is the biomass of these populations, and have they declined by an order of magnitude? Discards below the quota seems like an indicator of closure success rather than an indicator of a population’s conservation status.

Line 492-494

Remove inaccurate concluding sentence, or restate that extending the duration of the closures may reduce by-catch by capturing a greater portion of the period when cod and yellowtail populations occupy Georges Bank.

Line 495-496

“To be maximally effective…”

Line 496-497

Remove. Without a pre-post comparison the impact on the discard risk is unknown. Alternatively, have not … eliminated the discard risk.

Line 497-499

Therefore, because these populations have not achieved conservation targets …

Line 521-525

This information is crucial to interpreting the discard data presented in the results and should not be withheld until the final sentence of the discussion. For example, if the effort from pre to post closure reduced by 50%, then the 10-fold reduction in discard weight is a 5-fold reduction in discard rate per unit of effort.

Line 527-529

Remove “conservation impact”, or rephrase as “the degree of discarding unaffected by the current closure practices”

Line 528-529

“establishment of a dedicated to monitoring program”

Line 529-531

Remove “the spatial and temporal extent of these closures were insufficient to protect either species during spawning and” or rephrases as “insufficient to eliminate by-catch”

Line 531

replace “significant” with “consequential” or the like

Line 532-533

What is the evidence for this?

6. PLOS authors have the option to publish the peer review history of their article (what does this mean?). If published, this will include your full peer review and any attached files.

Reviewer #1: No

Reviewer #2: Yes: Jonathan W. Cummings

---

## [Author Response · Author response to Decision Letter 0]

28 Jul 2020

Response Attached. Thanks to the editor and reviewers for their thoughtful comments on our original submission, we hope these revisions address all of you comments and suggestions.

---

## [Decision Letter · Decision Letter 1]

9 Sep 2020

PONE-D-20-11298R1

Evaluating socio-economic and conservation impacts of management: A case study of time-area closures on Georges Bank

PLOS ONE

Dear Dr. Keith,

Thank you for submitting your manuscript to PLOS ONE. After careful consideration, we feel that it has merit but does not fully meet PLOS ONE’s publication criteria as it currently stands. Therefore, we invite you to submit a revised version of the manuscript that addresses the points raised during the review process.

Thank you for your thorough revision! Please see the comments from reviewer 1 suggesting minor changes, repeated below. Once you have addressed these two minor points, please return the manuscript, and I will be able to accept it for publication. 

"Page 18, Lines 413-414 - Rephrase this sentence to clarify that the *conservation* objectives may not be met by the closures. Perhaps a simple change like this will do: "We found that time-area closures had minimal impact on the scallop fishery, *however*, results suggest that the *conservation* objectives of the closure are not being fully realized.

Page 19, Lines 435-437 - Consider rephrasing this sentence as well. Should "ineffective" be "effective"? I think this paragraph could start with something simple, like this: "A lack of direct monitoring data poses challenges to evaluating the effectiveness of time-area closures" because monitoring data is important for evaluating socio-economic and conservation objectives."

We look forward to receiving your revised manuscript.

Kind regards,

Ulrike Gertrud Munderloh, Ph.D.

Academic Editor

PLOS ONE

Reviewers' comments:

Reviewer's Responses to Questions

**Comments to the Author**

1. If the authors have adequately addressed your comments raised in a previous round of review and you feel that this manuscript is now acceptable for publication, you may indicate that here to bypass the “Comments to the Author” section, enter your conflict of interest statement in the “Confidential to Editor” section, and submit your "Accept" recommendation.

Reviewer #1: All comments have been addressed

Reviewer #2: All comments have been addressed

2. Is the manuscript technically sound, and do the data support the conclusions?

Reviewer #1: Yes

Reviewer #2: Yes

3. Has the statistical analysis been performed appropriately and rigorously? 

Reviewer #1: Yes

Reviewer #2: Yes

4. Have the authors made all data underlying the findings in their manuscript fully available?

Reviewer #1: Yes

Reviewer #2: No

5. Is the manuscript presented in an intelligible fashion and written in standard English?

Reviewer #1: Yes

Reviewer #2: Yes

6. Review Comments to the Author

Reviewer #1: The authors have done an excellent job addressing reviewer comments! I have nothing substantive to add, just a couple minor suggestions:

Page 18, Lines 413-414 - Rephrase this sentence to clarify that the *conservation* objectives may not be met by the closures. Perhaps a simple change like this will do: "We found that time-area closures had minimal impact on the scallop fishery, *however*, results suggest that the *conservation* objectives of the closure are not being fully realized.

Page 19, Lines 435-437 - Consider rephrasing this sentence as well. Should "ineffective" be "effective"? I think this paragraph could start with something simple, like this: "A lack of direct monitoring data poses challenges to evaluating the effectiveness of time-area closures" because monitoring data is important for evaluating socio-economic and conservation objectives.

Reviewer #2: The authors have responded to my concerns and I can confidently respond affirmatively to the PLOS One review criteria (apart from the data availability question with is due to sharing restrictions). I thank the authors for the completeness of their responses, in particular to the nuanced methodical and results questions. Thank you for carefully addressing the semantic confusion around conservation objectives in the original manuscript submission. I hope that will avoid confusion for future readers and broaden the audience the manuscript reaches. If the authors are not familiar with value of information analyses (e.g. Runge et al 2011 Which uncertainty? Using expert elicitation and expected value of information to design an adaptive program) I encourage them to explore this methodology and consider it for future evaluations of this or other similar data for us in a management context.

7. PLOS authors have the option to publish the peer review history of their article (what does this mean?). If published, this will include your full peer review and any attached files.

Reviewer #1: No

Reviewer #2: **Yes: **Jonathan W. Cummings

---

## [Author Response · Author response to Decision Letter 1]

10 Sep 2020

Thanks to both reviewers for their helpful comments throughout. The two minor suggestions from Reviewer 1 have been incorporated into this draft.

---

## [Editor Report · Decision Letter 2]

24 Sep 2020

Evaluating socio-economic and conservation impacts of management: A case study of time-area closures on Georges Bank

PONE-D-20-11298R2

Dear Dr. Keith,

We’re pleased to inform you that your manuscript has been judged scientifically suitable for publication and will be formally accepted for publication once it meets all outstanding technical requirements.

Kind regards,

Ulrike Gertrud Munderloh, Ph.D.

Academic Editor

PLOS ONE
---

## [Editor Report · Acceptance letter]

28 Sep 2020

PONE-D-20-11298R2 

Evaluating socio-economic and conservation impacts of management: A case study of time-area closures on Georges Bank 

Dear Dr. Keith:

I'm pleased to inform you that your manuscript has been deemed suitable for publication in PLOS ONE. Congratulations! Your manuscript is now with our production department. 

Kind regards, 

on behalf of

Dr. Ulrike Gertrud Munderloh 

Academic Editor

PLOS ONE